# ERPs in Children and Adolescents with Generalized Anxiety Disorder: Before and after an Intervention Program

**DOI:** 10.3390/brainsci12091174

**Published:** 2022-09-01

**Authors:** Nikolaos C. Zygouris, Filippos Vlachos, Georgios I. Stamoulis

**Affiliations:** 1Department of Computer Science and Telecommunications, University of Thessaly, 35131 Lamia, Greece; 2Special Education Department, University of Thessaly, 35221 Volos, Greece; 3Department of Electrical and Computer Engineering, University of Thessaly, 38334 Volos, Greece

**Keywords:** children and adolescents with anxiety, event related potentials, CBT program

## Abstract

According to DSM 5, generalized anxiety disorder (GAD) is characterized by excessive, uncontrollable worry about various topics that occupies the majority of the subject’s time for a period of at least six months. The aforementioned state causes distress and/or functional impairments. This paper presents the outcomes of a pilot study that evaluated the implementation of cognitive behavioral therapy (CBT) and CBT with an SSRIs intervention program. The participants comprised 16 children and adolescents with GAD (8 males and 8 females) matched with 16 typically developing peers (8 males and 8 females) aged from 10 to 16 years old (M = 12.56 SD = 2.18). Baseline assessment consisted of event related potentials (ERPs), which indicated that participants with GAD presented cognitive deficits in attention and memory, as they exhibited longer P300 latencies. Following treatment with the CBT program and/or medication, children and adolescents with GAD did not present statistically significantly longer P300 latencies and reaction times in comparison to the control group. Lastly, children and adolescents who followed the CBT program or the CBT program with psychopharmacological assistance did not reveal statistically significant differences in 13 out of 15 topographic brain areas and in reaction time.

## 1. Introduction

In DSM 5 [1], generalized anxiety disorder (GAD) is characterized by excessive, uncontrollable worry about various topics that occupies the majority of the subject’s time for a period of at least six months. The worry causes distress and/or functional impairments and is associated with at least three of the following features: restlessness or feeling keyed up on the edge; being easily fatigued; having difficulty in concentrating or having one’s mind go blank; and exhibit irritability, muscle tension, and sleep disturbances. GAD is one of the more common anxiety disorders and occurs in anywhere from 5% to 20% of the child population [2].

Studies using the reappraisal paradigm tend to show prefrontal hypoactivation and increased prefrontal cortex amygdala connectivity [3,4]. Thus, the most prevalent pathophysiological model postulates that the disorder is caused by abnormal activation along a circuit encompassing the prefrontal cortex, the basal ganglia, and the thalamus, resulting in a disinhibition of abnormal maladaptive habits, over which patients are unable to exert sufficient cognitive control [5].

Many studies (e.g., [6,7]) on subjects with anxiety disorder indicate that this condition reduces the ability to sustain and focus attention while performing a task. Furthermore, performance can be easily distracted by other stimuli that are not related to the task, which implies that anxiety disorders decrease attention span and increase attention switching [8]. Additionally, anxiety plays an important role in cognitive abilities. An excessive reaction to stimuli and concentration difficulties are the most common symptoms of anxiety disorders. Cognitive accounts of anxiety suggest that information processing biases play a central role in the etiology and persistence of anxiety disorders [9]. Impaired performance can be compensated for by increased effort and the involvement of additional cognitive functions, and such results were presented in several electrophysiological studies. [6,7]. Furthermore, people with anxiety disorders face difficulties in the correct classification of information in the cognitive framework that they are called upon to cope with [10].

Women are found to be two- or three-times as prone to GAD criteria on a lifelong basis as men [11]. Both sexes exhibit differentiated symptoms, as females experience higher physical strain, fatigue, and muscle tension compared to males [12]. According to McLean et al.’s theory, on account of the fact that women are more susceptible to acquiring internalizing disorders compared to men, social and sex-related aspects may play an integral role in the display of physical symptoms [11]. Negative effects and neuroticism may be more prevalent in females than in males, and these characteristics have been linked to an increased risk of developing anxiety disorders in general and GAD in particular [13]. Women are also found to be more susceptible to anxiety, due to their reproduction-related nature. For instance, adaptive behavioral differences in childrearing appear to include stronger social cognition and the ability to tune into others in females, which are crucial for the cognitive and social development of children [14]. Women are more likely to be negatively affected by rejection, criticism, and separation, in light of the sex disparities described above, and these are integral elements of melancholy and anxiety disorders [15]. The NCS-R and CPES are two of the studies reporting a major sex discrepancy in rates of prevalence, according to which females have a higher percentage of GAD compared to males (for review, [16]).

The sex differences described in anxiety disorders are thought to be caused by a variety of factors, including genetic, neurodevelopmental, environmental, and neurobiological aspects. Striking variations between men and women in both the anatomy and function of the brain can be spotted in regions associated with anxiety, such as the prefrontal cortex, hippocampus, and extended amygdala complex [17]. In the neurobiology of anxiety disorders, female reproductive hormones, particularly estrogen and progesterone, may also play a significant role [18]. Their significant CNS modulating effects impact the way anxiety disorders are exhibited in females, the way they evolve, and the degree of response to treatment [19]. It is crucial to understand that sex differences in behaviors and coping mechanisms are relative rather than absolute, and that one sex just exhibits them more frequently than the other. The magnitude of these sex disparities is typically quite minor [16].

Event related potentials (ERPs) are an electrophysiological technique that is non-invasive and may could lead to the accumulation of information regarding brain activity related to the processing of cognitive information. It has been suggested that ERPs are the most widely studied method for obtaining correlations between brain electrophysiological activity and the dynamic processes of cognitive stimuli [20]. A well-studied component of ERPs is the P300 that has been suggested to reflect higher-order cognitive information processing and attentional functions associated with contextual evaluation when attending to the provided stimulus [21].

More precisely, it is thought that P300 latency reflects higher-order cognitive processes, including classification and stimuli appraisal [13]. Therefore, it has been proposed that P300 latency might operate as a temporal indicator of brain activity, underpinning the rapidity of attention allocation and immediate memory processes [14,15,22,23].

Several studies have used ERPs as a method of assessing anxiety disorders. A number of studies reported a relation between anxiety disorders and P300 components (e.g., [24,25,26,27]). Specifically, patients with anxiety disorders presented significantly longer latencies in P300 in comparison to the healthy controls [25,26,27,28,29,30,31]. However, there are studies that reported that the P300 latency was shorter in patients with anxiety in comparison to the control group [32,33]. It was suggested that the enlargement/reduction was most associated with fear stimuli, but not neutral or pleasant stimuli. It is speculated that attention bias for fear-related or novelty stimuli can account for the P300 changes in patients affected by anxiety spectrum disorders [34]. Furthermore, other anxiety related states such as impulsivity alters the P300 components. Moreover, several biological and demographic correlates (especially age) have been documented to have significant effects on P300 latency and amplitude [35]. Drug-free patients display longer P300 latencies, according to a recent review conducted by Sahoo [36].

The use of neuroscience to grapple with cognition, behavior, and environmental impact greatly contributes to the evolution and assessment of focused treatments and clinical practice protocols [37]. Brain plasticity, which is a relatively rapid and reversible change in brain function and structure, is expressed in learning, memory, and perception. Learning and memory are reflected at a cellular level by changes in synaptic plasticity. In other words, neurons are able to modulate the strength and structure of their interconnections as a result of experience [38,39].

Cognitive behavioral therapy (CBT) has a long been used for interventions whose goal is to improve behavioral, cognitive, and health outcomes. According to CBT models, processing is guided by cognitive frameworks/concepts that largely determine how information is attended to, interpreted, and remembered, as schemas in anxious individuals that are thought to be biased towards threat [40].

The FRIENDS (feeling worried, relax and feel good, inner thoughts, explore plans, nice work so reward yourself, don’t forget to practice, stay calm), is a family based cognitive behavioral treatment program [41,42,43], which is noted by WHO [44] as exhibiting strong evidence of being effective as a school-based intervention of anxiety.

The main aim of the present pilot research protocol was to compare the electrophysiological brain activity of children and adolescents with GAD (diagnosed in a state diagnostic center) using ERPs, with those of typically developing counterparts. The ERPs waveform utilized was the P300 latency. Subsequently, the group of children and adolescents with GAD participated in a remediation program using principals of the FRIENDS program or both the FRIENDS program and psychopharmacological assistance, and their electrical brain activity was reassessed.

Based on the aforementioned studies (e.g., [30,31]) the first hypothesis of the present study was that Greek children and adolescents that had already been diagnosed with GAD, would present significant longer P300 latency and reaction times in comparison to the control group. Based on relevant findings in studies with adults (e.g., [45]), the second hypothesis of the present study was that after the implementation of the intervention program, participants with GAD would present shorter P300 latency and reaction time scores compared to both typically developing peers and the same (GAD) participants pre-treatment. The third hypothesis of the present research protocol was that children with GAD post-treatment would present differentiated P300 latency and reaction time scores, according to the intervention program that was implemented.

## 2. Materials and Methods

### 2.1. Participants

Sixteen right-handed children and adolescents (8 males and 8 females) aged from 10 to 16 years old (M = 12.56 SD = 2.18), who had been diagnosed for GAD by children and adolescent Psychiatric hospital of Athens, participated in the present study. It is worth mentioning that none of the participants in the anxiety group had ever followed any intervention program, as they were assessed and diagnosed for the first time. The control group consisted of sixteen right-handed children and adolescents (8 males and 8 females) matched for age (M = 12.65 SD = 2.38) with participants with GAD. None of the participants that formed the control group had any psychopathological disorder. In the control and clinical groups, none of the participants had learning disabilities, developmental disorder, or significant visual or auditory impairments, according to their medical reports from their schools. All participants were recruited after reading informative newspaper articles, notifications regarding research inside hospital, and attending informative school meetings. It is worth mentioning that participants with GAD were diagnosed following standard and typical assessment protocols, such as a clinical interview using K-SADS, which was administered according to DSM 5 [1] by a child and adolescent psychiatrist, as well as using their answers to well-studied clinical tests, such as the self-report Revised Children’s Manifest Anxiety Scale (RCMAS) [46]. Finally, all human data included in this manuscript were obtained in compliance with the Helsinki Declaration and the guidelines of the Ethics and Deontology Committee of the University of Thessaly.

### 2.2. Electrophysiological Assessment

The P300 latency was recorded from 15 electrode sites (FP1, FPz, FP2, F3, Fz, F4, F7, F8, C3, Cz, C4, P3, Pz, P4, Oz) according to the 10–20 International System [47] using Ag-AgCl electrodes. All channels were referenced to linked mastoids, and the ground electrode was located at the nasion.

Using a Medronic computer system, (710 Medtronic Parkway, Minneapolis, MN, 55432-5604, USA) with amplifiers with a band range of 0.1–30 Hz, recordings and stimulation were conducted. EBNeuro software (Via P. Fanfani, 97/A—50127 Firenze—Italy) was employed. The electrode impedance was never more than 5 K. With a baseline of 200 ms prior to the stimulus and 800 ms after it, the sampling duration was 4 ms over a 1000-ms timespan. The auditory stimuli included tones of 1000 and 2000 Hz delivered binaurally through headphones at 75 dB SPL for 40 ms (10 ms rise and 10 ms fall period), with a 1500-ms interstimulus interval. The frequent stimuli (1000 Hz) were shown 140 times, while the oddball stimuli (2000 Hz) were provided 60 times (probability 0.30) (probability 0.70). The non-target stimuli consisted of the frequent stimuli. All children were seated in a comfortable reclining armchair. Eye blinks were monitored with electrodes located above the right eye; nevertheless, all children were advised to have their eyes closed.

Additionally, the P300 component’s latency was assessed at all derivations, for both standard and target stimuli. The P300 component was the longest positive peak that followed the N200 waveform within a delay range of 250–450 ms [48]. Trials were excluded from analysis if their voltage exceeded around 70 μV in any of the 15 channels (except EOG) or if they corresponded to error responses. The averaging of periods was carried out separately for target and non-target stimuli, after applying a baseline correction for the 200 ms pre-stimulus interval. It was a priori determined that only children with at least 30 artifact free periods for non-target and target stimuli would be included in the analysis [49,50,51,52,53]. The assessment was the same for all children participating in the post-treatment program. Furthermore, all participants were instructed to press a button as quickly and as accurately as possible following the identification of the oddball (target) stimulus, using their right hand, and the reaction times were recorded. The experiments were performed in a shielded room, free from noise and interruptions.

### 2.3. Intervention Program

All sixteen children diagnosed with GAD underwent the intervention program, which was provided for a period of three months (10 sessions), while two additional sessions, which were conducted one and three months, respectively, following the completion of the treatment. Children’s sessions were implemented one day per week and lasted from 45 min to one hour. All sessions were implemented using the basic principles of the family based cognitive behavioral therapy program FRIENDS. Specifically, children’s sessions aimed at building a strong relation with their body, in order to recognize the physical symptoms that indicate anxiety, to become friends with themselves, use cognitive techniques in order to deal with anxiety, reward themselves for trying hard, build friendships, construct an autonomous social support network, and discuss with their friends when they are in difficult and stressful situations.

The FRIENDS program also incorporates family skills components. All parents of children with GAD formed a group and underwent 10 sessions, one day per week, separately from their children, which lasted from 45 min to one hour. The parent’s sessions aimed at recognizing and dealing appropriately with their own anxiety; training in reinforcement strategies, including logical rewards for gradually facing fear situations; instruction in planned ignoring, in order to deal with children’s excessive complaints; included role play strategies with examples of their child’s fearful behaviors, including cognitive techniques and problem solving skills; as well as developing a support group among parents [44].

It must it highlighted that those 8 children (4 male and 4 female) who followed the FRIENDS program used psychopharmacological assistance, serotonin reuptake inhibitor (SSRIs), following the instructions of the child and adolescent psychiatrist that diagnosed them. It must be taken into consideration that the results collected by this pilot study will be used by a larger study that is in progress, to evaluate the intervention of the selected CBT program.

## 3. Data Analysis

The feasibility outcomes are reported using summary statistics. Data analyses were performed using SPSS 25.0 (IBM Corp. Released 2016. IBM SPSS Statistics for Windows, Version 24.0. Armonk, NY, USA: IBM Corp.) Pre- and post-treatment analyses were conducted with Bonferroni post hoc analysis and mean confidence intervals were examined to assess the difference between baseline and follow-up outcomes. Cohen’s d-effect size [54] is also reported. Furthermore, the comparison between the participants that followed only the FRIENDS program and those that followed the psychotherapy along with medication was calculated using non-parametrical statistical analysis and the effect size was not conducted. Finally, one way ANOVA analysis was performed, so as to trace the differences between participants of both sexes with GAD, both prior to the intervention program and after.

## 4. Results

Descriptive statistics were utilized, in order to obtain mean scores and standard deviations of P300 latency, which was recorded from 15 scalp locations. Bonferroni post hoc test was used to compare the P300 latency of children and adolescents with GAD pre- and post-treatment and the control group. The statistical significance threshold in this study was established at 0.01, due to the large number of statistical tests conducted with few individuals, which decreased the likelihood of making Type I statistical errors (i.e., incorrectly rejecting the null hypothesis). Additionally, Cohen’s d effect sizes were computed and shown. Effect sizes below 0.20 are regarded as modest, between 0.20 and 0.50 as medium, and larger than 0.50 as large in magnitude, according to Cohen [54]. Table 1 presents mean scores and standard deviations of P300 latency in all 15 brain areas of children and adolescents that participated in control group and children and adolescents diagnosed with GAD before and after treatment.

Bonferroni post hoc analysis revealed that children and adolescents with GAD pre-treatment had statistically significantly (*p* < 0.01) longer P300 latencies in all 15 brain regions compared to their average peers that participated in the control group. The effect-sizes calculated for the comparisons of the brain activity in the fifteen electroencephalographic sites were all large, ranging from 0.91 to 0.74. The effect size suggests that the differences in P300 latencies between the two groups were large and clinically important.

Next, the analysis focused exclusively on the children and adolescents with GAD, to compare their scores in post-treatment and, additionally, seeking to determine the degree of their electrophysiological improvement. Following the implementation of the treatment program the same comparisons were performed between the two groups.

Bonferroni post hoc analysis indicated that no statistically significant differences between the control group and children and adolescents with GAD post-treatment (*p* > 0.05) were detected in all electroencephalographic sites examined. A careful analysis of the computed effect-sizes revealed that, in all recorded brain areas, the differences between the two groups ranged from 0.00 to 0.33, which was most definitely not the case in the pre-treatment measurements.

While the P300 latency was being recorded, participants underwent a behavioral measurement test, as they had to press the button on the joystick when they perceived the oddball stimuli. Table 2 upper panel presents the mean scores and standard deviations of reaction time for the control group and children and adolescents with GAD pre-treatment. Moreover, the upper part of Table 2 presents the statistical significance of the reaction time between children with GAD pre-treatment and the control group. The bottom part of Table 2 presents mean scores, standard deviations, and the significance of the reaction time between children and adolescents that participated in the control group and children and adolescents with GAD post-treatment.

As is presented in the upper part of Table 2, children and adolescents with GAD had statistically significantly (*p* < 0.01) longer reaction times when they perceived the oddball stimuli in comparison to children that participated in the control group. Furthermore, a large effect size (0.66) was observed, as is presented by Cohen’s d [54].

Following the implementation of the intervention program, and as can be inferred from the bottom part of Table 3, the children’s and adolescents’ reaction times in both groups did not differ significantly. In addition, the effect size (0.12) that was calculated was considered small.

Next, the analysis focused on examining possible differentiations of the P300 latencies of the children and adolescents with GAD, according to their participation in the intervention program either by only implementing FRIENDS CBT or by implementing both FRIENDS and SSRIs. It must be mentioned that a Mann–Whitney non-parametrical statistical analysis was calculated, as 8 children and adolescents with GAD underwent the FRIENDS CBT and 8 children and adolescents followed both FRIENDS and SSRIs medication. The results are presented in Table 3. Table 3 presents mean scores, standard deviations, and statistical significance of the P300 latency of participants with GAD, according to which intervention program they followed. It must be highlighted that because of the use of a non-parametrical statistical analysis, the effect size was not calculated.

As is presented in Table 3, participants with GAD did not present statistically significant differences in 13 out of 15 topographic brain areas, according to the intervention program that was followed. However, children and adolescents that underwent only the cognitive behavioral psychotherapeutic program presented significantly longer P300 latencies in two central and parietal brain topographic areas (Cz and Pz) in comparison to children that followed the psychotherapeutic program with medication.

Furthermore, a Mann–Whitney non-parametrical statistical analysis was calculated, in order to examine the reaction time of children and adolescents with GAD, according to the intervention program that was implemented. No statistically significant difference was found in the reaction time (U = 0.083) dependences to the intervention program that was followed.

Finally, it must be pointed out that one way ANOVA was employed, so as to identify differences in P300 latency between male and female participants with GAD in both pre- and post-treatment. According to the findings of the analysis, the female participants displayed larger P300 latencies, but compared to the male participants there was no significant statistical difference (*p* > 0.05).

## 5. Discussion

The present pilot study was conducted in order to compare the electrophysiological brain activity of children and adolescents with GAD using ERPs with those of typically developing counterparts. The first hypothesis, which was based on previous studies [17,18,19,20,21,22], compared the P300 latencies and reaction times of children and adolescents with GAD with those obtained from a control group. The expected longer latencies in the P300 component in participants with GAD was indicated by the present study.

The longer latency of P300 waveform that was presented by children and adolescents with GAD in the present research and other studies possibly explains the cognitive deficits that are present in children and adolescents with anxiety disorders (e.g., [9]). Moreover, the most frequent cognitive components of anxiety are attention and memory for information processing disorders [55]. Since the latency of P300 reflects the promptness of stimulus assessment and demonstrates the efficiency of cognitive functioning, it can uncover the aforementioned disabilities [7]. Studies using P300 latency suggest that participants with impaired memory present significantly prolonged auditory P300 latency [21]. Furthermore, it has been shown that a prolonged latency of P300 indicates disorders in attention [56]. In general, longer P300 latencies indicate inferior mental performance [48].

The second hypothesis of the present study arose from studies of brain plasticity and neural cortical activation, which performed CBT and/or SSRIs intervention in children, adolescents, and adults with anxiety disorders and found differences in P300 latencies and reaction times. In the present study, children and adolescents with GAD presented shorter latencies in the P300 component and reaction times, after the implementation of the intervention program and did not present statistical differences in P300 latency and reaction time in comparison to their typically developed peers; a result that verifies our second hypothesis. According to a number of research works (for review, [50,57]) evaluating school-based anxiety prevention programs, those based on CBT have produced effective outcomes and shown notable post-intervention decreases in anxiety, as evaluated by a number of anxiety tests. Furthermore, other studies using the P300 component in patients with obsessive-compulsive disorder and serotonin reuptake inhibitor treatment reported that the improvement in cognitive processing due to pharmacological treatment modified the P300 component, bringing it close to the controls. It is also proposed that an abnormal P300 indicates that the participant is not cognitively processing the auditory evoked stimulus appropriately [45,58,59].

Additionally, regarding neurobiological aspects, the present findings can be explained by neurotransmitters that are both involved in anxiety disorders and in the generation of the ERPs component. Gamma-amino butyric acid (GABA) has long been regarded as the center of regulation of anxiety. In addition, 5-HT (serotonin) plays an important role in the development of anxiety disorders. A large number of studies suggest that when 5-HT concentration is increased, it is positively correlated with the increase of anxiety symptomatology [60]. Dopaminergic neurotransmission is widely referred to as a neurotransmitter that controls fear and anxiety [61]. The same neurotransmitters are responsible for the presence of the P300 functional components of ERPs [62].

The above results suggest that children and adolescents with GAD strengthen their attentional and working memory abilities, as they are measured electrophysiologically by the P300 latency, and behaviorally by attentional modification, as measured by reaction time [63].

The third hypothesis of the present study was focused only on the group of children and adolescents with GAD that participated in a remediation program using the principals of the FRIENDS program or the FRIENDS program with psychopharmacological assistance; and the hypothesis that their electrical brain activity and reaction time would differ was reassessed and, to a large extent, rejected. Non-parametric statistical analysis revealed that there was no significant difference between the groups in 13 out of 15 brain regions and in reaction times. Consequently, children and adolescents with GAD post-treatment did not present altered P300 latency and reaction time scores according to the intervention program. The only differences were found to central and parietal brain topographic regions, which might reflect neurotransmitters such as serotonin, which plays a central role in both anxiety and P300.

A fourth statistical analysis was conducted in order to reveal possible differences between the sex of GAD participants. In more detail, a notable sex difference was supported, with the women consistently reporting to have an increased rate of GAD in comparison to men [11]. However, the findings of the present pilot study suggest that females present greater P300 latency in comparison to males, with no statistically different findings.

The major strength of the present pilot study may be that, according to the best of our knowledge, there is a lack of research protocols about children and adolescents at such an early age (M = 12.56 SD = 2.18). Moreover, a small number of studies reported that the participants were drug-free before the electrophysiological assessment of their brain activity and none of them mentions possible psychotherapeutic interventions. It is worth keeping in mind that, in the present research protocol, all participants had never been diagnosed by a public hospital and none of them had ever participated in any rehabilitation (pharmacotherapeutic and/or psychotherapeutic) program. The aforementioned differences in designing the present study might also explain the results that were produced by the pre-treatment and post-treatment procedure.

In reviewing the findings of the present research protocol, some limitations should be considered. Not all of the ERP waveforms described in the literature were used in the current study. As it is the most commonly used in relation to higher mental ability issues, only P300 was selected. Another weakness is the limited sample size, which might have reduced the statistical analysis power (16 children and adolescents participated in the control group and 16 children and adolescents with GAD). The strength of our study is defined by the large number of statistically significant outcomes that are given. Another limitation is that the control group did not have a P300 follow-up examination. However, the P300 potential has been shown to be reliable, exhibiting no significant changes over time periods of two months to two years [64]. In more detail, it is suggested that despite maturational changes in late adolescents, the P300 latency shows significant stability and score agreement over a 2-year span, accounting for 55% of the total variance, and has virtually the same reliability as the split-half measurement. The normal variation of P300 latency that is attributable to subject state factors is not sufficient to mask the normal range of inter-subject variability among adolescents [65]. It must be taken into consideration that the reliability of P300 latency is quite consistently lower in older age groups [66]. Owing to the three months intervention program and the small age of the participants, the control group of the present study was not reassessed.

In terms of behavior, the results of the present study, which reported longer latencies for children and adolescents with GAD in comparison to the control group, indicate that children and adolescents with anxiety disorders face attentional and memory deficits, which may explain their low academic performance. Since research data suggests that the increased P300 latency implies attention [67] and memory deficits [68] of people with anxiety disorders, the improvement of the P300 latency that was demonstrated in the present research protocol indicates an improvement of the higher cognitive functions of the participants with GAD. In addition, the improvement in reaction times noted by children with GAD after the intervention program may be an indication of an improvement in their cognitive behavior and might also be an indication of improved organization. These findings can be taken into consideration in the implementation of intervention programs in schools. 

## 6. Conclusions

Despite these drawbacks and limitations, the findings of the current pilot research are encouraging, both for the evaluation of children and adolescents with GAD and for the ongoing global effort of the scientific community to develop and implement better remediation programs for these children. The outcomes of a remediation program on a very small sample were used to inform the present pilot project; however, further research is required. Nevertheless, it is worth pointing out that the results acquired in the present pilot protocol can be considered as having practical implications for children and adolescents with GAD, since a specific CBT intervention program that changed participants’ electroencephalographic activity was carried out. Additionally, stressful situations and negative emotions such as anxiety can interfere in school performance. Attentional control theory, developed by Eysenck et al. [8], describes anxiety as a disrupter of our ability to control our attention, which means that we are more readily distracted while performing a task. Attentional control theory therefore implies that the cognitive capacity is reduced and efficiency is weakened, irrespective of whether the stimuli prompts are internal (stressful thoughts) or external (duties) To conclude, intervention programs focusing on emotional and cognitive factors must be implemented, in order to increase school performance. Future studies might focus on reproducing this CBT intervention in broader populations of children and adolescents with GAD, to enable a sound generalization. More significantly, such research studies should, wherever feasible, be longitudinal, meaning they should include long-term follow-up evaluations of the children who received remediation. Such a research strategy might result in more comprehensive data and, thus, more reliable findings, leading to the creation of evidence-based interventions intended particularly for children and adolescents, and potentially applicable to all young people dealing with anxiety spectrum disorders.

## Figures and Tables

**Table 1 brainsci-12-01174-t001:** The mean scores, standard deviations of P300 latency from all recorded brain areas from children and adolescents that participated in the control group, and children and adolescents that were diagnosed with GAD pre- and post-treatment.

Electro/ Encephalographic Sites	Control Group		GAD Pre-Treatment		GAD Post-Treatment	
	M	SD	M	SD	M	SD
FP_1_	303.35 ms	11.95 ms	352.68 ms	17.91 ms	297.48 ms	9.85 ms
FP_z_,	305.93 ms	10.98 ms	356.80 ms	16.90 ms	303.15 ms	8.56 ms
FP_2_,	311.73 ms	8.68 ms	365.75 ms	16.68 ms	307.22 ms	9.02 ms
F_3_	307.28 ms	11.76 ms	355.25 ms	14.70 ms	304.22 ms	9.99 ms
Fz	308.38 ms	11.97 ms	359.07 ms	14.10 ms	307.77 ms	9.66 ms
F_4_	317.58 ms	12.53 ms	367.24 ms	19.13 ms	312.76 ms	9.24 ms
F_7_	310.90 ms	12.26 ms	358.17 ms	12.23 ms	306.73 ms	11.47 ms
F_8_	319.29 ms	11.41 ms	370.63 ms	15.65 ms	314.71 ms	11.66 ms
C_3_	321.09 ms	11.56 ms	356.26 ms	18.17 ms	323.16 ms	9.20 ms
Cz	324.99 ms	10.54 ms	364.82 ms	12.33 ms	326.33 ms	9.04 ms
C4	329.22 ms	11.48 ms	368.80 ms	14.54 ms	330.33 ms	10.39 ms
P_3_	318.44 ms	11.98 ms	359.59 ms	18.68 ms	321.98 ms	7.11 ms
Pz	326.93 ms	12.06 ms	369.04 ms	12.72 ms	328.50 ms	8.45 ms
P_4_	329.30 ms	8.05 ms	376.11 ms	16.53 ms	329.81 ms	7.03 ms
Oz	328.60 ms	12.69 ms	369.41 ms	11.33 ms	325.93 ms	12.82 ms

**Table 2 brainsci-12-01174-t002:** Mean scores, standard deviations, significance, and effect size of children and adolescents that formed the control group, and children and adolescents with GAD pre- and post-treatment reaction time. Statistical significance was calculated.

Task	Control Group		GAD Pre-Treatment		
	Mean	SD	Mean	SD	F
Reaction Time	321.01 ms	28.28 ms	396.72 ms	35.78 ms	39.35 **
Task	Control Group		GAD Post-treatment		
	Mean	SD	Mean	SD	F
Reaction Time	321.01 ms	28.28 ms	324.29 ms	35.78 ms	0.061

Note ** *p* < 0.01.

**Table 3 brainsci-12-01174-t003:** Mean scores, standard deviations, and significance of P300 latency in participants with GAD that followed only the cognitive behavioral psychotherapeutic program and children and adolescents with GAD that underwent cognitive behavioral psychotherapeutic program along with SSRIs medication. Statistical significance of all brain areas that were recoded.

Electro/ Encephalographic Sites	Groups				
	Participants that followed only FRIENDS Program		Participants that followed FRIENDS Program and Medication		
	Mean	S.D.	Mean	S.D.	U
FP1	298.81 ms	5.08 ms	296.14 ms	13.32 ms	0.279
FPz	303.63 ms	4.31 ms	302.66 ms	12.36 ms	0.505
FP2	308.09 ms	10.41 ms	306.35 ms	7.45 ms	0.721
F3	306.49 ms	4.87 ms	301.94 ms	13.35 ms	0.234
Fz	309.69 ms	3.94 ms	305.86 ms	13.26 ms	0.195
F4	315.35 ms	8.89 ms	310.17 ms	11.53 ms	0.442
F7	306.96 ms	13.58 ms	306.49 ms	9.49 ms	0.721
F8	315.90 ms	11.78 ms	313.52 ms	12.20 ms	0.442
C3	325.81 ms	11.37 ms	320.51 ms	6.01 ms	0.234
Cz	329.58 ms	10.60 ms	323.09 ms	4.22 ms	0.038 *
C4	335.19 ms	11.80 ms	325.46 ms	6.16 ms	0.083
P3	325.19 ms	7.98 ms	318.77 ms	4.59 ms	0.195
Pz	333.66 ms	7.70 ms	323.34 ms	6.44 ms	0.021 *
P4	333.26 ms	6.01 ms	326.47 ms	6.58 ms	0.105
Oz	327.01 ms	9.91 ms	324.87 ms	15.84 ms	0.798

* *p* < 0.05.

## Data Availability

All data generated or analyzed in this study are included in the study.

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
