# Peer review of "ERPs in Children and Adolescents with Generalized Anxiety Disorder: Before and after an Intervention Program"

_brainsci, 2022, doi:10.3390/brainsci12091174_

Round 1

Reviewer 1 Report

Thank you for the opportunity to review this study entitled “ERPs in children and adolescents with Generalized Anxiety Disorder: before and after an intervention program.” (brainsci-1852592).

The study aimed at assessing a the implementation Cognitive Behavioural Therapy (CBT) and CBT with psychopharmacological assistance (SSRIs intervention program).

In my opinion, the research topic is relevant, and the study is interesting. Parallelly, there are some issues that need to be addressed before the paper will be suitable for publication.

1.     Abstract: the information about the sample should be deepened (Mean age and SD? Percentage of men and women?) to provide a clear picture of what will be presented in the paper.

2.     Method: please, provide more information about the recruitment of the samples.

3.     A “Data analysis” section should be added, and the first few lines of the “Results” part should be moved to this section.

4.     In the “Conclusions” section the authors reported interesting implications for future research. However, I believe that this section should also be enriched with the practical implications of the results obtained in this study.

In general, I enjoyed this paper. In my opinion, after the authors make small changes, it will be ready to be published.

Best wishes

Author Response

Thank you very much for the helpful changes that you addressed to our work.

  1. According to your first comment we made changes in Abstract (lines 16-18)
  2. According your second commend we mage changes in methodology (lines 164-166).
  3. We have included a data analysis section in the paper (lines 241-251)
  4. We have enriched the conclusions section.

Reviewer 2 Report

line 188 change to Cohen's

Author(s) should further explain possible reasons for no significant differences after treatments. Null findings should not be ignored and attempts to explicate are helpful.

Author Response

Thank you very much for the helpful changes that you addressed to our work.

  1. The text was checked by a native English speaker, and we corrected syntactical, grammatical, and semantical problems according to the reviewer’s suggestion.
  2. We have further explained reasons for no statistical differences and made attempts to explicate null findings.

Reviewer 3 Report

The manuscript of Zygouris et al., subjected to pilot study of ERPs in children and adolescents with Generalized Anxiety Disorder before and after CBT and pharmacological interventions presents accurately performed and in some ways pioneering work. The manuscript generally well written however some considerations need to be addressed prior to publication, as follows:

 Major: The authors used participants of two sexes in the study, but did not discuss the potential contribution of sex both to the development of GAD and possible outcomes of interventions. It is necessary to clarify whether gender differences could influence the results obtained.

 Minors:  

11)  The first sentence in the abstract and the first sentence in the introduction duplicate each other.

22)   The term "gender" in the abstract (line 16) is more correct to replace with "sex"

33) It is not fully understood why authors referred to work focused on OCD when described the pathophysiological model of GAD (p.1, lines 35-39). Actually, OCD is not related to anxiety disorders in DSM-5.

44) The text on p.2, lines 76-82, sounds like a trivia and could be eliminated without loss to the meaning and structure of the introduction.

55) The description of statistical methods is needed to be transferred from results to materials and methods section.

66)  Given that table 1 and figure 1 are depict the same data, authors need to choose one type of data representation. If authors will prefer the graphical representation they should presented SD and statistical significance on the bar plots.

77)  Page 6, lines 236 and 241. Obviously it's not about table 3 but about table 2.

Author Response

Thank you very much for the helpful changes that you addressed to our work.      

  1. We have clarified gender differences (lines 55-83 (Introduction section), lines 250-252 (data analysis section), lines 248-352 (results) lines 414-418 in discussion)
  2. We have changed the sentence in the abstract (lines 10-11)
  3. We did not refer to the term “gender”
  4. We found a better reference in explaining the pathophysiological model of GAD (line 42)
  5. We have included a data analysis section after materials and methods and before results in the paper (lines 241-251).
  6. We have chosen the table as a more detailed representation of the results.
  7. We have changed the number of the table 2.

Round 2

Reviewer 2 Report

I have reviewed the revised manuscript and it meets my concerns.